Using visuo-kinetic virtual reality to induce illusory spinal movement: the MoOVi Illusion

Harvie Daniel S. d.harvie@griffith.edu.au 1
Smith Ross T. 2
Hunter Estin V. 1
Davis Miles G. 2
Sterling Michele 1
Moseley G. Lorimer 3 4
1 Recover Injury Research Centre, Centre of Research Excellence in Road Traffic Injury, Menzies Health Institute QLD, Griffith University, Griffith University , Gold Coast , Australia
2 Wearable Computer Lab, University of South Australia , Adelaide , Australia
3 Sansom Institute for Health Research, University of South Australia , Adelaide , Australia
4 Neuroscience Research Australia , Sydney , Australia
Fadiga Luciano
Electronic publication date: 2017 Feb 22
Publication date: 2017
Volume: 5
Electronic Location ID: e3023
Received 2016 Nov 10; Accepted 2017 Jan 24
Copyright: ©2017 Harvie et al.
Copyright year: 2017
Copyright holder: Harvie et al.
License: This is an open access article distributed under the terms of the Creative Commons Attribution License, which permits unrestricted use, distribution, reproduction and adaptation in any medium and for any purpose provided that it is properly attributed. For attribution, the original author(s), title, publication source (PeerJ) and either DOI or URL of the article must be cited.
License URL: https://creativecommons.org/licenses/by/4.0/

Keywords: Chronic pain, Virtual reality, Bodily illusion, Brain training, Central mechanisms

Funding: Physiotherapy Research Foundation Griffith University New Researcher Grant National Health and Medical Research Council of Australia This project is supported by the Physiotherapy Research Foundation and Griffith University New Researcher Grant. GLM is supported by a Principal Research Fellowship from the National Health and Medical Research Council of Australia. The funders had no role in study design, data collection and analysis, decision to publish, or preparation of the manuscript.

==============================
Background

Illusions that alter perception of the body provide novel opportunities to target brain-based contributions to problems such as persistent pain. One example of this, mirror therapy, uses vision to augment perceived movement of a painful limb to treat pain. Since mirrors can’t be used to induce augmented neck or other spinal movement, we aimed to test whether such an illusion could be achieved using virtual reality, in advance of testing its potential therapeutic benefit. We hypothesised that perceived head rotation would depend on visually suggested movement.

Method

In a within-subjects repeated measures experiment, 24 healthy volunteers performed neck movements to 50o of rotation, while a virtual reality system delivered corresponding visual feedback that was offset by a factor of 50%–200%—the Motor Offset Visual Illusion (MoOVi)—thus simulating more or less movement than that actually occurring. At 50o of real-world head rotation, participants pointed in the direction that they perceived they were facing. The discrepancy between actual and perceived direction was measured and compared between conditions. The impact of including multisensory (auditory and visual) feedback, the presence of a virtual body reference, and the use of 360o immersive virtual reality with and without three-dimensional properties, was also investigated.

Results

Perception of head movement was dependent on visual-kinaesthetic feedback (p = 0.001, partial eta squared = 0.17). That is, altered visual feedback caused a kinaesthetic drift in the direction of the visually suggested movement. The magnitude of the drift was not moderated by secondary variables such as the addition of illusory auditory feedback, the presence of a virtual body reference, or three-dimensionality of the scene.

Discussion

Virtual reality can be used to augment perceived movement and body position, such that one can perform a small movement, yet perceive a large one. The MoOVi technique tested here has clear potential for assessment and therapy of people with spinal pain.

Introduction

The relationship between the brain’s representations of the body and the integrity of the body itself is complex. In both recovery from an acute injury and in the transition from acute to chronic pain, resolution of tissue pathology and resolution of pain do not go hand in hand. One explanation for ongoing pain after tissue healing is that the brain retains neural encoding consistent with a body that is still threatened—the brain misrepresents the body. This threatened state then continues to drive redundant protective responses such as pain. Thus, chronic pain could be viewed as a mismatch between the body and its neural representation. This theme is common to several contemporary theories of chronic pain ((see cite for examples and review of such theories—the cortical body matrix paradigm (Moseley, Gallace & Spence, 2012), the Imprecision Hypothesis of chronic pain (Moseley & Vlaeyen, 2015), Explain Pain (Moseley & Butler, 2015)).

Illusions that distort bodily perception confirm that divergence between real and perceived bodily state is possible, and that the brain’s representation of the body can be artificially manipulated. The flagship example of this is the rubber hand illusion (RHI). To perform the RHI, the participant’s hand is hidden, and a rubber hand is placed in view. The experimenter simultaneously strokes the out-of-view real hand, and the in-view rubber hand. Within a minute, most participants report feeling touch coming from the rubber hand, to which they also report a sense of agency (Botvinick & Cohen, 1998). Further, when asked to point to their hand, a shift in perceived limb position towards the rubber hand is apparent: a kinaesthetic drift (Botvinick & Cohen, 1998). This illusion is possible by the integration of seen and felt sensory inputs and engages a range of physiological and perceptual processes (Moseley et al., 2008; Barnsley et al., 2011; Moseley, Gallace & Spence, 2012).

Mirror therapy is an effective treatment for phantom limb pain and may have application in other types of pathological hand pain (Chan et al., 2007; Cacchio et al., 2009; Ramachandran & Altschuler, 2009; Bowering et al., 2013). It appears to utilise a similar illusory principle to that observed for the RHI. In mirror therapy, the subject places the affected limb behind a mirror and observes the moving reflection of the healthy limb. The subject concurrently performs a partial or imagined movement of the painful or missing limb. The integration of motor commands, with proprioceptive and visual information, results in an illusory perception of movement that aligns with the movement of the mirrored hand. This augmented perception may be viewed as one of a brain convinced that the affected limb is in a state similar to that of the unaffected side—present, free moving, and healthy—which may counter neural encoding that continues to hold the body as threatened.

When effective, illusions such as mirror therapy must act via brain-based mechanisms. Indeed, pathological hand and phantom limb pain are associated with signs of central dysfunction, including aberrant perception of size and shape of the body part, and functional changes in the somatosensory cortex (Marinus et al., 2011). Evidence consistent with distorted cortical body representation in chronic pain, however, is not unique to painful limbs. For example, sensory changes and functional reorganization of the primary somatosensory cortex has been observed not only in pathological hand pain (Juottonen et al., 2002; Di Pietro et al., 2013; Di Pietro et al., 2015), carpal tunnel syndrome (Tecchio et al., 2002), and phantom limb pain (Flor et al., 1995), but also in spinal pain conditions (Flor et al., 1997; Catley et al., 2014). The extent of this reorganization is associated with the characteristics of pain including intensity (Flor et al., 1995) and duration (Flor et al., 1997), suggesting their relevance to clinical symptoms.

Given the potential contribution of the brain to ongoing spinal pain, cortical representations could be a valuable target for intervention. This prospect is particularly relevant for a number of reasons. Firstly, spinal pain—specifically neck and lower back pain—together form the single greatest contribution to years lived with disability world-wide (Vos et al., 2015). Secondly, patho-anatomical theories of spinal pain have so far failed to explain ongoing pain, or lead to reliable treatments (Waddell, 2004; O’Sullivan, 2012). Brain-based treatments, including interventions employing illusions, have been suggested as a way to directly target cortical representations (Senkowski & Heinz, 2016; Moseley & Flor, 2012; Moseley, Gallace & Spence, 2012; Wand et al., 2011; Boesch et al., 2016; Wallwork et al., 2016). It is suggested that illusory interventions, such as mirror therapy, might assist by altering the way the painful body part is encoded in the brain (Ramachandran & Altschuler, 2009). Because of the practical constraints of mirror therapy—which relies on an unaffected and duplicated body part such as a hand or foot—it cannot be used for spinal pain. Creating a perception of a healthy and free moving spine, however, might be achieved using virtual reality.

Recent developments in virtual reality technology suggest that it can be used to alter perception of the body in space. A virtual reality concept known as ‘re-directed walking’ uses altered visual feedback to manipulate perceived orientation, by shifting the virtual-world in ways that are not aligned with real-world movement (Steinicke et al., 2010). The goal of this is to induce corrections in real-world movement that, for example, enable a user to walk in a continuous straight line in a virtual world, while remaining within constraints of real-world environments—where they are walking in a circle (Steinicke et al., 2010). Preliminary tests suggest that participants can be made to walk in a circle with a diameter as small as 22 metres, while being convinced they are walking in a straight line. While re-directed walking pertains to whole body movement through space, researchers have also developed ways of augmenting visual feedback relating to motion of individual body segments within the body (Azmandian et al., 2016; Kokkinara, Slater & López-Moliner, 2015; Harvie et al., 2015; Harvie et al., 2016). For example, Azmandian et al. (2016) showed that manipulating virtual arm movement can induce corresponding corrections in actual limb movement. Such examples of interactions between visual input and motion, that highlight the relationship between vision and bodily perception/kinesthesia, can be traced back to Helmholtz (1910), who’s subjects mis-pointed at targets when wearing prism glasses that displaced the visual field to one side. Our own research provides some evidence that such illusory virtual movement may modulate chronic pain, by demonstrating that altered virtual neck motion changes the relationship between real-world movement and pain, in people with persistent neck pain (Harvie et al., 2015). While this is likely due to the capacity of vision to alter perceived motion of the neck, the relationship between visual feedback and perceived neck motion has not been investigated. Understanding the relationship between visual feedback and perceived neck motion (or other spine motion) may be important for developing therapeutic applications for chronic pain based on altered visual-kinesthetic feedback.

In this study we aimed to establish whether altered visual-kinaesthetic feedback could change perceived neck movement. We hypothesised that perceived neck movement would be dependent on altered visual-kinaesthetic feedback.

Methods

Subjects

Twenty-four participants (14 males, age(SD) = 29(8)), were recruited through flyers placed around the University Campus. Participants were informed that we were testing how virtual-reality affects body perception, but they were not informed that the visual feedback would be systematically altered. Participants were excluded if they had current pain or history of chronic pain. Based on a consistent effect seen in piloting, we expected a large effect size for the main effect (ηp2=0.14). Based on the conventional 80% power, we estimated a sample of 10 people was required. To improve sensitivity to detect the effect of additional variables such as augmented auditory feedback, this sample was increased a priori to 24. All participants gave written informed consent. The protocol was approved by the Institutional Ethics Committee of Griffith University (Protocol Number: 2016/378).

Equipment and software

A virtual-reality (VR) technique known as redirected walking modulates visual-kinaesthetic feedback by tracking real-world movement and then feeding this back into the virtual environment in an understated or overstated form. This illusion is typically applied to the whole body moving through space—to enable a user to walk continuously in a virtual world, while in the spatial restraints of an indoor environment (Steinicke et al., 2010). We have previously modified this illusion to be applied to individual joints or body segments relative to the rest of the body (Harvie et al., 2015; Harvie et al., 2016). We call this the Motor-Offset Visual Illusion (MoOVi) because the user real world motor command is offset by given factor and then translated to a virtual world that can under or overstate true movement. The factor by which real-world movement is translated to virtual movement is known as the gain value. We used a VR head-mounted display (HMD) designed for immersive VR environments (Oculus Rift DK2; Oculus VR, Irvine, CA). Custom built MoOVi software was used to map six scenes (captured with 360° photography) to an immersive virtual reality environment with three degrees of freedom (DOF), and to apply the selected gain values (to visualize the MoOVi illusion see https://youtu.be/2Lu7G0cge2Q).

Measurement of kinesthetic drift

Participants were positioned in a well-supported chair that restricted torso movement. Participants were asked to rotate only their head to a specific marker placed within the virtual world, to stop, and to point in the direction that they felt they were facing (see Fig. 1). The digital marker in the virtual world changed position depending on the gain value applied, such that it always corresponded to 50° degrees of real-world movement. A digital camera placed directly above the participant, captured the participant’s response. Markers were placed on the axis of shoulder rotation, and the base of the index finger. The direction of perceived head rotation in each trial was then extracted from screen shots using Kinovea video analysis software, by placing digital markers in the location of the physical makers. Kinovea has previously shown to be highly reliable for angular measures (Balsalobre-Fernández et al., 2014).

Figure 1 Change in perceived body position (C) was operationalized as the difference between perceived head rotation (B) after movement during altered visual feedback, relative to during movement with normal visual feedback (A).

Multisensory feedback

The principle manipulation was visual-feedback that under or overstated the amount of rotation being performed by the participant. During neck movements to 50° of rotation, the virtual reality system administered corresponding visual-kinaesthetic feedback that was offset by a factor of 50%–200% (simulating between half and double the amount of movement actually occurring). Real-world rotation to 50° was kept constant to enabled relative control of proprioceptive and vestibular inputs, such that the independent effects of the MoOVi illusion could be examined. To test whether various additional multisensory and contextual factors influenced the strength of the kinesthetic illusion, we divided the study into two parts.

Part one contained three multisensory conditions: 1. Vision only, 2. Vision + Sound, and, 3. Vision + Sound + Avatar. In the Vision only condition, only visual feedback was provided. In the Vision + sound condition, synchronous spatially defined auditory feedback was also provided. The auditory feedback had spatial properties through manipulation of left–right ear synchronicity and volume, which increased or decreased, relative to defined coordinates in the virtual world. In the Vision + Sound + Avatar condition, a virtual body was included, to act as a virtual anchor for the real body, such that the altered virtual head movement would be presented as a relative to (a visual representation) of the rest of their body.

Part two of the study contained three contextual conditions designed to alter scene familiarity/sense of immersion, and the 360° photography based immersive virtual reality vs. a modelled three-dimensional scene. These conditions were the: 1. Familiar Scene, constructed from a 360° photograph of the lab in which the experiment took place, 2. Unfamiliar Scene (360° scene), and 3. Three-dimensional Scene (the same office scene in 3D). The 360° immersive virtual realities allow interactive viewing of panoramic photos in a spherical view. The three-dimensional office scene provided a more complete virtual representation by imparting dimensional and locational information for objects within the scene (i.e., table and chairs) allowing stereo perception and improved perception of depth relative to the 360° photographic scenes.

Experimental design

We used a within-subjects, repeated-measures design. In Part one, four gain settings were used: 50%, 100%, 150% and 200%. All trials within each gain setting were undertaken in a single block so as to minimize the likelihood of participants realizing our manipulation, and to minimize the risk of motion sickness. The order of conditions was counterbalanced between participants. Within each block, three sensory conditions were presented in randomized order. These conditions were: 1. Vision only, 2. Vision + Sound, and, 3. Vision + Sound + Avatar. Each condition was presented twice within each block, so that each block contained six trials. This resulted in a total of 24 trials (i.e., four gain settings x three multisensory conditions x two repetitions). In Part two, two different gain values were used: 100% and 180% to reduce the time burden. All trials at each gain value were undertaken in a single block. The order of blocks, was counterbalanced between participants. Within each block, three contextual conditions were presented in random order. These conditions were: 1. Familiar Scene, 2. Unfamiliar Scene, 3. Three-dimensional Scene. These conditions enabled us to determine whether familiarity and dimensional cues could alter the strength of illusion and thus the primary outcome of perceived movement. Each condition was presented twice within each block, so that each block contained six trials. This resulted in a total of 12 trials (i.e., two settings x three contextual conditions x two repetitions).

Manipulation checks

During the experiment, participants were asked to rotate their head to the right, to stop, and to then physically point in the direction of perceived head orientation. We suspected that the real-time shift in perceived head orientation during rotation might be diluted through the post-movement task of actively considering their orientation, because internal reference frames, based on proprioceptive and vestibular information, might have greater time to compete with the bogus visual information. Thus, at the end of the experiment, participants were asked to turn their head to the minimum and maximum real-world movement that they perceived across the trials. By comparing these head rotation angles to actual head movement (always 50°), and to the pointing data, we aimed to gauge potential bias in the pointing data, and obtain a secondary measure of the overall magnitude of kinaesthetic drift.

Statistical treatment

In order to test the main hypothesis that visual information that overstates or understates true rotation alters perceived movement, we compared perceived head orientation between visual feedback conditions, using repeated-measures ANOVA with Bonferroni-corrected pairwise comparisons. To test for interactions with multisensory and contextual/dimensional variables, these factors were entered as covariates. Prior to analysis, perceived orientation in each condition was converted to a difference score in degrees, relative to their average perceived head orientation across conditions. Alpha was set at p = 0.05. The magnitude of the effects were quantified using partial eta squared (ηp2) and interpreted with respect to Cohen’s guidelines (0.01 = small, 0.059 = medium and 0.138 = large (Cohen, 1988). Based on the hypothesis that individual factors might influence the effect of conflicting sensory information on perceived neck movement, we also conducted exploratory analyses to determine if the effect related to gender or simulator sickness susceptibility. To do this we calculated the slope coefficient created by the increasing perceived head orientation angles, at the four increasing rotation gain settings, for each participant. To test whether the effect related to gender we performed an Independent Samples T-test comparing the slope coefficient in males vs. females. To test whether the effect related to susceptibility to motion sickness, we performed a Pearson’s correlation analysis between simulator sickness score and the slope coefficient representing the effect.

Results

Participants

Forteen males and 10 females participated. Average age of participants was 30 years (range: 20–42). Following the experiment, participants reported only mild motion sickness symptoms: mean (range) Simulator Sickness Questionnaire = 8% (0–38%). Only one participant scored over 20%, and most participants reporting a very mild sense of nausea.

Kinaesthetic drift

A linear relationship was observed between change in visual-feedback and change in perceived movement (see Fig. 2). The overall effect of altered visual feedback was 7.5(6.4)° as measured using the pointing task. Visual feedback that overstated real-world movement by a factor of 2, resulted in 4.6(5)° drift in perceived orientation, whereas understating real-wold movement by a factor of .5, resulted in on average −2.9(3.3)° drift in perceived orientation (see Table 1 for results by condition and gain factor). The repeated measures ANOVA for Part 1 confirmed a large effect of altered sensory feedback (F(3, 92) = 6.4, p = 0.001, ηp2=0.17) on perception of head movement (Fig. 2). That is, altered visual feedback caused a proprioceptive drift in the direction of the visually suggested movement. The magnitude of the drift was not moderated by the addition of illusory auditory feedback, nor by the presence of the avatar (F(3, 92) = 1.5, p = 0.2, ηp2=0.05). The repeated measures ANOVA for Part 2 also revealed a large effect of altered sensory feedback (F(1, 94) = 16.5, p < 0.001, ηp2=0.15) on perception of head movement, confirming the findings of Part 1. The magnitude of the drift was not moderated by scene familiarity or three-dimensionality (F(1, 94) = 0.2, p = 0.7, ηp2<0.01).

Figure 2 Perceived direction of head orientation, relative to average perceived direction across conditions.

Table 1 Change in perceived movement (Mean(SD)) for each gain factor and sensory condition as measured by the pointing task.

	Gain factor	
Sensory condition—Part 1	0.5	1	1.5	2	
Vision	−1.8(3.7)	1.4(2.4)	2.1(2.6)	5.1(4.8)	
Vision + Avatar	−3.8(3.0)	−1.0(1.9)	1.5(2.0)	4.2(4.0)	
Vision + Avatar + Audio	−2.9(3.1)	−0.5(2.7)	1.8(3.3)	4.0(5.6)	
Vision + Audio	−3.2(3.2)	0.0(2.1)	2.7(4.6)	5.0(5.4)	
Overall	−2.9(3.3)	0.0(2.4)	2.0(3.2)	4.6(5.0)	
Sensory condition—Part 2	1	1.8			
3D	0.2(1.6)	3.6(3.6)			
Familiar	−0.2(1.8)	3.6(4.4)			
Unfamiliar	0.2(1.5)	3.9(4.3)			
Overall	0.0(1.6)	3.5(3.8)			

Manipulation check: perceived head orientation

When asked to reproduce the range of head orientation angles performed while in the virtual world, participants on average perceived between 31° (SD 7, range 21–45) and 65° (SD 13, range 47–97), despite always turning to 50°. Paired t-test confirmed the statistical difference between the smallest and the largest perceived head rotation (p < 0.001; mean difference =33° (SD 12, range 18–70)).

Exploratory analysis

The slope coefficient representing the effect also did not differ by gender (t(22) − 1.7, p = .105), although a trend towards a greater effect in females was noted (Slope coefficient; Males =1.9(1.3), Females =3.2(2.3)). No correlation was shown between simulator sickness scores and tendency to experience greater kinaesthetic drift (p = .6, r =  − .11).

Discussion

We aimed to establish whether altered kinaesthetic feedback could change perceived neck movement. We hypothesised that perceived neck movement would be dependent on altered visual-kinaesthetic feedback. Our hypothesis was confirmed, as evidenced by main effect of gain value on perceived orientation and that perceived orientation followed gain value in a systematic way. The effectiveness of the illusion seems to be unaffected by the addition of cross-modal (auditory) input, nor by scene familiarity. While the kinaesthetic drift demonstrated by the pointing task was modest, the manipulation check suggested that this action captured only a portion of the perceptual shift. This is analogous to the drift in perceived location of the hand during the rubber hand illusion, where asking participants with closed eyes to point to where they feel their hand to be, captures only a portion of the real-time perceptual shift (Botvinick & Cohen, 1998). The current findings demonstrate that virtual reality can induce illusory neck movement analogous to the body-illusion underlying mirror therapy for pathological and phantom limb pain. The finding is consistent with other virtual reality studies showing that altered visual feedback can manipulate whole body (Steinicke et al., 2010) and limb (Kokkinara, Slater & López-Moliner, 2015; Azmandian et al., 2016) movement.

The motor offset visual illusion (MoOVi)

Perceptual models explain visual illusions by the way that perceptions are derived from inferences about the world, derived from past experience and multisensory information that is uncertain and sometimes conflicting (Cheng et al., 2007). In the current example, perception of body orientation would be informed most notably by: 1. Expected movement, ‘the efferent copy of the motor command’, 2. Proprioceptive and vestibular feedback, and 3. The altered visual-kinaesthetic feedback. Thus, the resulting perception of head orientation is a best guess inference based on the relative influence of these variables. As demonstrated by other bodily illusions, visual information is weighted heavily and therefore shows a particular propensity to influence perception. Perceptual models explain this by suggesting that the most reliable sensory channel is weighted most heavily and, with respect to spatial data, vision is the most reliable sensory channel (Cheng et al., 2007). This idea is also consistent with the principle that the influence of one neural network on another depends in part on its precision (see cite for relevant reviews Nicolelis & Lebedev, 2009; Wallwork et al., 2016; Wallwork, Bellan & GL, in press).

While the magnitude of change in perceived head rotation, as measured by the primary outcome, does not appear consistent with the heavy weighting of visual information, our manipulation check suggests that the actual manipulation of perceived movement was much greater. A different methodology may be more capable of objectifying the change in perceived head orientation (see Limitations and future directions). Nonetheless, the magnitude of the effect is consistent with similar measures of kinaesthetic drift previously used, and which suggest such measures capture only a portion of the change in body position experienced during the illusion. For example, despite the convincingness of the rubber hand illusion, participants show a kinaesthetic drift of only 23 mm towards the rubber hand, when measured by a pointing task (Botvinick & Cohen, 1998). In contrast, observing real-time changes movement behaviour in response to altered feedback may be more a more sensitive measure. For example, Kokkinara, Slater & López-Moliner (2015) observed up to 22° of drift in arm movement in response to modulation of virtual arm movement.

Illusory movement as therapy? Plausible mechanisms

To date, persistent pain treatments targeting suspected tissue pathology have shown to have very limited or no ongoing benefit (see e.g., Van Tulder, Koes & Malmivaara, 2006; Van Tulder et al., 2006). The quest for better treatments has seen attention shift towards targeting central mechanisms (Wand et al., 2011; Moseley & Flor, 2012). While cognitive and behavioural approaches targeting thoughts, beliefs and behaviors undoubtedly have potential to alter neural processes associated with pain, considering brain-based approaches from a perceptual and brain science perspective will likely lead to a range of different treatment avenues. For example, recent perspectives highlight several pathways by which perception-altering associative learning processes might lead to chronic pain (Moseley & Vlaeyen, 2015; Tabor et al., 2017; Zaman et al., 2015). It follows that similar mechanisms applied differently might reverse some of these central processes and reduce pain. In this vein, recent reviews have highlighted a potential role for multisensory illusions in developing future treatments (Senkowski & Heinz, 2016; Moseley & Flor, 2012; Boesch et al., 2016).

The view of chronic pain employed here, is one where pain persists because the affected body part(s) in someway continues to be represented in the brain as being under threat and requiring certain protections—perhaps because the injurious event-related encoding persists (Moseley & Vlaeyen, 2015). One way to consider altering this threatening body-related neural encoding is to disconfirm implicit expectations of pain, such as those associated with movements. This requires an experience that is normally painful to be experienced without pain. Such violations of expectation are known to be powerful drivers of learning (Rescorla & Wagner, 1972), and are thought to alter prior implicit expectancies that may drive the resulting pain responses (Tabor et al., in press).

Under normal clinical conditions, creating pain-free or non-threatening movement experiences, through which to extinguish implicit expectations of pain, can be difficult. The MoOVi illusion presents an opportunity for users to experience ranges of pain-free movement that exceed the expected range of pain-free movement. Doing so may assist to reinstate a less threatened bodily representation. Indeed we have created a Samsung Galaxy Gear application that patients can use at home that aims to serve this aim. The app instructs a rotation exercises and progressively exaggerates these movements over a number of repetitions, while limiting real-world movement to within pain-free limits through a calibration process. We are shortly commencing initial clinical testing of this application for the treatment of chronic neck pain, where central mechanisms are thought to play a role (See Australia and New Zealand Trial Registry, Universal Trial Number: U1111-1190-8259). This is conceptually similar, not only to mirror therapy, but to Graded Motor Imagery (GMI)—which targets progressive activation of motor processes without triggering unwanted protective responses (Moseley, 2004; Moseley, 2006). Indeed the final phase of three in graded motor imagery is illusory movement with mirror therapy, which has preliminary evidence of benefit for arm pain (Bowering et al., 2013). The current results raise the possibility that the MoOVi illusion might allow a graded motor imagery-like approach to neck and other spinal pain conditions.

Limitations and future directions

As mentioned, the change in perceived movement due to the bogus visual feedback was considerably less than we expected. However, our manipulation check suggested the primary outcome was diluted and further research should explore better ways to assess the magnitude of the illusion and its effect on motor output. For example, we suspect that in the process of reaching out, the illusion may have been partially broken because the expectation of seeing the arm in the visual field would have been violated. The use of a virtual arm, or a reaching task below the visual field may be a more appropriate measure. While auditory and contextual cues did not appear to modulate the strength of the illusion in this study, it is possible that our measure was simply too insensitive to detect its effect. Supporting this thesis, is a recent study showing that, that in general, a greater number of additional sensory cues—including vibrotactile, wind and sound—corresponds to improved performance and immersiveness in virtual environments (Feng, Arindam & Lindeman, 2016).

While the current study is framed with respect to chronic pain, its assessment and treatment, this is clearly not a clinical study. The potential benefit is speculative and further research is required before taking this to a clinical phase. A further direction for future study, is examining individual factors that relate to susceptibility to this kinaesthetic illusion. For example, sensitivity to interoceptive information appears to modulate the degree of kinaesthetic drift induced by the rubber hand illusion (Tsakiris, Tajadura-Jiménez & Costantini, 2011). In the therapeutic context, identifying factors that predict sensitivity to the illusion may be relevant in predicting treatment efficacy. While gender and susceptibility to simulator sickness did not significantly predict effectiveness of the illusion, future research might investigate whether other individual factors modulate the effect.

Conclusion

Virtual reality can be used to augment perceived movement and body position, such that one can perform small real-world movements, yet experience a larger movement. The current study extends this finding to movement of the neck. Thus, the MoOVi technique may have similar central effects and therapeutic potential to treatments currently only applied to the limbs, for example mirror therapy.

Supplemental Information

Supplemental Information 1 Open data

Click here for additional data file.

Additional Information and Declarations

Competing Interests

Author Contributions

Human Ethics

Data Availability

GLM has received support from Pfizer, Kaiser Permanente USA, Agile Physiotherapy, Results Physiotherapy, the Port Adelaide Football Club, workers’ compensation boards in Australia, North America and Europe. GLM receives royalties for books on pain and rehabilitation and speaker fees for lectures on pain and rehabilitation.

Daniel S. Harvie conceived and designed the experiments, performed the experiments, analyzed the data, contributed reagents/materials/analysis tools, wrote the paper, prepared figures and/or tables, reviewed drafts of the paper.

Ross T. Smith and Miles G. Davis conceived and designed the experiments, contributed reagents/materials/analysis tools, reviewed drafts of the paper.

Estin V. Hunter conceived and designed the experiments, performed the experiments, reviewed drafts of the paper.

Michele Sterling contributed reagents/materials/analysis tools, reviewed drafts of the paper.

G. Lorimer Moseley wrote the paper, reviewed drafts of the paper.

The following information was supplied relating to ethical approvals (i.e., approving body and any reference numbers):

The protocol was approved by the Institutional Ethics Committee of Griffith University (Protocol Number: 2016/378).

The following information was supplied regarding data availability:

Dataverse

doi:10.7910/DVN/94RA9O

https://dataverse.harvard.edu/dataverse/PainData.

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
