# Peer review of "Using visuo-kinetic virtual reality to induce illusory spinal movement: the MoOVi Illusion"

_PeerJ, doi:10.7717/peerj.3023_

## Round 0.1 · original submission · Minor Revisions

· Academic Editor

Minor Revisions

The paper has been found to be potentially very interesting. Please address the reviewer comments in your revision/rebuttal.

Reviewer 1 ·

Basic reporting

no comments

Experimental design

The authors aim is "to establish whether altered visual-kinaesthetic feedback could change perceived neck movement...". However the protocol is not dissociating trunk fem head movement. Indeed, to investigate neck movement (only proprioceptive input), 2 conditions should be tested (trunk movement with respect to head fixed vs head movement with respect to trunk fixed). In the present case the authors only investigate head movement, mixing vestibular, visual and neck proprioception. This is confusing all along the ms, see for instance in the Discussion (line 282) the notion of “spinal movement” that is ambiguous (head movement is not only a spinal movement but also eye movement).
The authors partly justify this study "to evaluate its potential likeness to mirror therapy". Because nothing in the protocol and the results section is related to this therapy I would delete this "noise" in the Introduction section.
A video (or at least several photos) of what the participants actually see in the virtual reality set up would be useful to the reader. Indeed, it is presently difficult to gauge the realism/familiarity of the artificial display.

Validity of the findings

Because the set up introduces a visuo-vestibular conflict, one can expect motion sickness. The authors quantify this using a questionnaire. However, verbal report to evaluate motion sickness is one possibility that relies on subjective/explicit measurement but does not quantify the implicit/non verbal physiological effect. This approximation leads to consider participants' responses as homogenous defined by a mean value and and SD. This a potential weakness of the study: Can the authors reconsider their results (fig2) by introducing the SSQ score of each subject in the calculation of the kinesthetic drift? I know that it is better for science to present invariants behaviors and consistent results. However in the frame of a therapeutic solution (that seems a serious concern for the authors, see above), one may accept the idea that each futur patient will not present the same amount of pain and will not present similar sensitivity to visuo-vestibular conflict and equal motion sickness effect. Differently, can the authors make subgroups inside the whole population tested that would take into account such singularity?
In the Discussion section (line 297 “This idea is also consistent with the principles that the influence of one neural network on another depends in part on its precision… ») seems a rather loosy assertion: which precision the authors speak about? How can we measure the precision of a neural network? Does it depend on the performance of the devices that regularly changes with technology improvement? This term is vague in the context of this present paper: the authors should define this term or delete.

Additional comments

After reading the Results one can expect a Discussion focusing on sensorimotor conflict and the role of vision in the particular sensory context of the present setup while the authors consistently discuss the chronic pain problem (line 319 to 338). This paragraph appears very long regarding the few material provided in the study. This part distracts the reader from missing arguments about sensorimotor conflict induces by the present protocol.

·

Basic reporting

The paper describes a virtual reality technique that gives participants false feedback about their neck movements.
It describes an experimental study with 24 healthy volunteers.
The underlying goal is the suggestion that this might be used for therapy with patients with spinal pain.

Overall I find that the paper is very good, and I have the following points for its improvement:

(1) I think it would help the reader to spell out more fully about how these findings might be used in the therapeutic context. Some possible examples, or previous case studies, or even anecdotal observations could be useful. What mechanisms might be involved in this.

(2) There is a vast literature on redirected walking and now these techniques are being applied to haptic feedback, where, for example, the arm movements of a participant are distorted in virtual reality so that they believe they are touching different objects, but in reality they are always touching the same one. Please see

Azmandian, M., M. Hancock, H. Benko, E. Ofek, and A.D. Wilson. 2016. Haptic Retargeting: Dynamic Repurposing of Passive Haptics for Enhanced Virtual Reality Experiences. In Proceedings of the 2016 CHI Conference on Human Factors in Computing Systems. ACM. 1968-1979.

https://www.researchgate.net/profile/Eyal_Ofek/publication/302073801_Haptic_Retargeting_Video_Showcase_Dynamic_Repurposing_of_Passive_Haptics_for_Enhanced_Virtu

al_Reality_Experience/links/579bd98308ae80bf6ea345a9.pdf

Also the gain idea discussed in this paper has also been explored in
Kokkinara, E., M. Slater, and J. López-Moliner. 2015. The Effects of Visuomotor Calibration to the Perceived Space and Body, through Embodiment in Immersive Virtual Reality. ACM Transactions on Applied Perception (TAP). 13:3.

(3) In the results section rather than diving straight away into F-tests it would be useful to have a paragraph that just describes what the data suggests in straightforward terms - referring to diagrams and summary statistics. Then the F-tests etc can be used to support the conclusions.

Experimental design

The experimental design is straightforward and I have no further comments on this.

Validity of the findings

The data is well-reported subject to my comment above.

Additional comments

As in the first section above.

·

Basic reporting

Line 113: Spinal pain should be further described at some point. Perhaps some examples of individuals for whom such a treatment might be appropriate for this type of therapy.

Line 137: it might be helpful to delineate the subgroups or at least what question they are trying to answer

Line 249: The statistical properties are well defined, but a table of the central tendencies of each condition would be helpful.

Line 257: the syntax “<0.00” seems a bit odd

Experimental design

Lines 142: Presumably the marker was in different places in the environment corresponding to the gain factor, was the speed of movement controlled?

Line 237: are cohen’s d reported in the manuscript?

Validity of the findings

Line 224: This seems like somewhat of a biased manipulation check. By asking them the smallest and largest deviation, it seems very likely that the smaller rotation will be less than the larger. Was there an individual who reported the largest deviation smaller than the smallest deviation?

Additional comments

This is a potential interesting manuscript which assesses the perception of head orientation when feedback is altered in a virtual environment. When the gain of visual rotation is altered over a 3 fold range, the reported direction of the head is also altered. The addition of various other sensations or immersions had little impact on the findings. These data are couched in terms of pain. Though this could be a potential application of this approach, a more balanced discussion of these findings in the context of the previous and current work in altered perception and reporting of joint angles would be appropriate. As written there seems to be a large portion of the literature not included in this manuscript. It is difficult to judge how meaningful this work is without this proper survey of the literature. For example, the introduction focuses closely on a narrow body of literature that does not align with data presented. Without data regarding the effect of this approach on pain, a wider focus is warranted throughout most of the introduction and the latter part of the discussion.

---

## Round 0.2 · Minor Revisions

· Academic Editor

Minor Revisions

Please, follow Reviewer #3 suggestions and resubmit asap.

Reviewer 1 ·

Basic reporting

No comment

Experimental design

no comment

Validity of the findings

no comment

Additional comments

After reading this new version I consider that this paper is now suitable for publication in peerj

·

Basic reporting

The paper is well written and satisfies well the requirements of the journal.

Experimental design

The paper satisfies the requirement of primary research with well-defines research questions, methods, and analysis.

Validity of the findings

The findings are valid.

Additional comments

The author has responded to all previous comments and adjusted the paper appropriately and it is now reader for publication.

·

Basic reporting

no comment

Experimental design

no comment

Validity of the findings

no comment

Additional comments

The broadened presentation of the manuscript is appreciated. The manuscript remains heavily focused on pain, this remains a bit odd as no individuals with pain are assessed and it remains unknown if such illusions can be observed in folks with spinal pain.

I am a bit surprised that virtual reality has not been used more often to induce illusionary movements. The fact the additional steps you took to immerse the individual (avatar, audio, familiarity) did not have an effect seems a bit more noteworthy than is currently made out to be. Have any of the other manuscripts in this area find that such immersion might not matter?

You may want to clarify 'degrees' when discussing the '360 immersive virtual reality' in the abstract

Steinike et al XXX should be corrected

[Mead(SD)] should be corrected

---

## Round 0.3 · accepted · Accept

· Academic Editor

Accept

Many compliments! Excellent work!

·

Basic reporting

no comment

Experimental design

no comment

Validity of the findings

no comment

Additional comments

I appreciate the thoughts!